# Sustainable Incorporation of *Chlamydomonas reinhardtii* Powder into Flour-Based Systems: Investigating Its Influence on Flour Pasting Properties, Dough Physical Properties, and Baked Product Quality

**DOI:** 10.3390/foods14213621

**Published:** 2025-10-24

**Authors:** Zhihe Yuan, Jiaojiao Wang, Shangyan Ma, Jianhui An, Longchen Shang, Yexing Tao, Cheng Tian, Lingli Deng

**Affiliations:** College of Biological and Food Engineering, Hubei Minzu University, Enshi 445000, China; 202430433@hbmzu.edu.cn (Z.Y.);

**Keywords:** *Chlamydomonas reinhardtii*, dough, soda crackers, cookies

## Abstract

*Chlamydomonas reinhardtii* serves as a sustainable and high-quality protein source. In this study, *Chlamydomonas reinhardtii* powder (CRP) was used to replace wheat flour at proportions of 0% (C0), 2.5% (C2.5), 5% (C5), 7.5% (C7.5), and 10% (C10) for the production of soda crackers and cookies. The study compared the pasting and farinographic properties of wheat flour with that of CRP and found that C2.5 demonstrated optimal performance in terms of stretching energy and stretching ratio. Regarding the expansion ratios of soda crackers and cookies, C2.5 achieved the highest value compared to C5, C7.5, C10, and had no significant difference compared to C0. The incorporation of CRP resulted in an increase in the *L** value and whiteness index (WI), a reduction in the chromaticity value (C*), and a noticeable shift towards a greener hue in the overall color. This study offers a thorough exploration and establishes a robust theoretical foundation for baked foods enriched with CRP.

## 1. Introduction

Soda crackers and cookies, as highly consumed baked goods, have gained significant popularity due to their affordability, ease of storage and portability, diverse varieties, and extended shelf life [1]. However, traditional recipes often exhibit relatively low nutritional complexity, with high levels of sugar, sodium, and food additives, while protein and dietary fiber contents are relatively low [2]. Due to the demand for heathier snacks, improving product nutritional value while maintaining quality has become a critical research focus. In recent years, microalgae, as an emerging protein source, have garnered increasing attention. Microalgae offer advantages such as short growth cycles, easily controlled cultivation conditions, and no competition with traditional agriculture for land resources, demonstrating substantial technological and commercial potential [3].

Microalgae exhibit potential as protein alternatives due to its high protein contents [4], balanced amino acid compositions, abundant vitamins, minerals, short-chain and long-chain polyunsaturated fatty acids, carotenoids, enzymes, and dietary fibers [5,6,7]. Beyond commonly studied microalgae, *Chlamydomonas reinhardtii* shows great potential as a novel food ingredient. With broad biological applications, *Chlamydomonas reinhardtii* serves as a model organism widely used in biofuel and bioproduct production [8]. *Chlamydomonas reinhardtii*, having gained approval as a novel food resource in China in 2022, has also been granted new food raw material usage licenses in countries such as the United States and Singapore [7]. Jin et al. [9] demonstrated that lactic acid bacteria-fermented *Chlamydomonas reinhardtii* powder (CRP) significantly enhances steamed bun quality. Khemiri, et al. [10] reported superior sensory scores for bread enriched with CRP. Despite this, applications of CRP in soda crackers or cookies remain limited. The inclusion of CRP undoubtedly improves the nutritional characteristics of baked goods. Jin, Gu and Zhang [9] confirmed that the nutritional value of CRP-enriched baked products increased significantly with CRP substitution. However, consumer acceptance of CRP exhibits certain limitations. Although few studies address CRP’s acceptability in soda crackers or cookies, CRP shares similarities with *Chlorella* in terms of flavor and sensory scores [11]. Sahni, et al. [12] and Batista, et al. [13] found no significant differences in sensory scores between cookies containing 6% *Chlorella* and control groups. This suggests that reasonable control of CRP addition ratios could broaden its market potential for soda crackers and cookies. While prior studies have validated the health benefits and consumer acceptance of CRP-enriched baked goods, investigations into CRP’s impact on wheat dough quality and its interaction mechanisms in soda crackers or cookies remain limited. Additionally, previous CRP studies predominantly focused on nutrition improvements at high substitution ratios, limiting practical market application potential.

In this study, CRP substituted low-gluten flour at ratios of 0% (C0), 2.5% (C2.5), 5% (C5), 7.5% (C7.5), and 10% (C10) to explore its effects on the pasting properties, the farinographic and extensographic properties, and the color, expansion ratio, and texture properties of soda crackers and cookies. The aim of this study is to develop CRP fertilized soda crackers and cookies with suitable CRP content. This study provides valuable data regarding the incorporation of CRP into soda crackers and cookies, which may inform future product development and market exploration.

## 2. Materials and Methods

### 2.1. Materials

The ingredients employed in the formulation of the *Chlamydomonas reinhardtii* powder-enriched soda crackers and cookies included the following: low-gluten wheat flour (LWF) (Xinxiang Xinliang Grain and Oil Processing Co., Ltd. Xinxiang, China), *Chlamydomonas reinhardtii* powder (CRP) (Tou Yun Biotechnology Co., Ltd. Changzhi, China), yeast (Angel Yeast Co., Ltd. Wuhan, China), butter (Shanghai Zheng Hong Food Co., Ltd. Shanghai, China), milk (Pineau Food Co., Ltd. Shanghai China); the baking soda, eggs, sugar and salt were purchased from local commercial retailers. In this study, CRP was mixed with LWF at ratios of 0%, 2.5%, 5%, 7.5%, and 10%. The specific nutritional composition content is shown in Table 1.

### 2.2. Pasting Properties

In the pasting property test, varying concentrations of CRP (0%, 2.5%, 5%, 7.5%, 10%) were incorporated into LWF to create five composite formulations designated as C0, C2.5, C5, C7.5, and C10. The pasting properties of these mixtures were quantitatively assessed using a Rapid Visco Analyzer (RVA-Eritm; PerkinElmer, Springfield, IL, USA) in accordance with established analytical procedures. For each test, 4.00 ± 0.01 g of the sample was precisely weighed into an aluminum sample canister, and 25.00 ± 0.05 mL of deionized water was added. The mixture was homogenized by rapidly stirring 10 times with a plastic paddle. Subsequently, the paddle was secured to the axial drive assembly with positional calibration to the sample receptacle. To mitigate sedimentation bias, the viscosity measurement protocol commenced within 1 min post-preparation. Real-time viscosity parameters were autonomously captured through the viscosity-time curve profiling via the instrument-linked computational system. All methodologies conformed to the American Association of Cereal Chemists (AACC) International Approved Method 76-21 [14].

### 2.3. The Physical Properties of Dough

#### 2.3.1. Farinographic Properties

The farinographic properties under varying CRP substitution levels (0%, 2.5%, 5%, 7.5%, and 10%) were investigated using a JFZD Farinograph (Dongfu Jiuheng Instrument Technology Co., Ltd., Beijing, China) with a 300 g mixing bowl, a torque-controlled system operating on the same principle as the Brabender Farinograph. For each trial, 300.0 ± 0.5 g of flour was weighed into the bowl and thermostatically controlled at 30.0 ± 0.2 °C. After preheating and dry mixing for 1 min, distilled water was titrated incrementally to achieve a target torque of 500 ± 20 FU, ensuring standardized dough consistency. Critical parameters were automatically derived from the farinograph curves. All measurements strictly complied with the AACC International Approved Method 54-21.

#### 2.3.2. Extensographic Properties

The extensographic properties under varying CRP substitution levels (0%, 2.5%, 5%, 7.5%, and 10%) were analyzed using a JMLD150 Extensograph (Dongfu Jiuheng Instrument Technology Co., Ltd., Beijing, China), a calibrated system functionally equivalent to the Brabender Extensograph. Dough specimens (150.0 ± 0.5 g) were conditioned at 30.0 ± 0.2 °C for fermentation periods of 45, 90, and 135 min. Following fermentation, samples were positioned in the extensograph trough, and uniaxial deformation was induced using an extensograph hook at a constant extension rate of 14.5 ± 0.5 mm/s. Three key parameters were measured: Stretching energy (cm^2^): Total area under the curve, representing the work required to rupture the dough. Extensibility (mm): Maximum displacement before dough rupture. Stretching resistance (BU): Peak force recorded during extension. Stretch ratio was calculated as the ratio of stretching resistance to extensibility:
(1)$$\mathrm{Stretch}\;\mathrm{Ratio}=\frac{\mathrm{Stretching}\;\mathrm{Resistance}\;(\mathrm{BU})}{\mathrm{Extensibility}\;(\mathrm{mm})}$$

Triplicate measurements were conducted for each fermentation interval to ensure methodological robustness. All measurements strictly complied with the AACC International Approved Method 54-10.

### 2.4. Baking Procedure of Soda Crackers and Cookies

The soda crackers and cookies were formulated by wheat flour and that with CRP, based on a total flour weight of 150 g. The detailed formulation and the key process parameters are presented in Table 2. The specific baking procedure is as follows:

Soda crackers: Baking soda and salt were incorporated into the CRP-LWF mixture. Following thorough blending, the yeast-milk mixture (prepared by fully dissolving yeast in pure milk) was introduced in three sequential additions. Melted butter was then added as the final ingredient. A dough mixer (AM-CG108-1, North American Electrical Co., Ltd., Zhuhai, China) was utilized to process the mixture into a shaggy consistency, which was subsequently kneaded into a cohesive dough ball. Dough proofing was conducted at room temperature for 30 min. A dough sheeter subsequently flattened the dough into a uniform sheet with a thickness of 2.5 mm. The dough sheet was precisely cut with a knife, yielding slices, each measuring 5 cm in length and 3 cm in width. These slices were then baked in an oven for 20 min. Reference Xie, et al. [15] methods.

Cookies: Sugar, salt, and melted butter were put into a mixing bowl, and mixed thoroughly until the ingredients were fully incorporated. Subsequently, the egg liquid was added and beaten until the mixture was homogenous. The CRP-LWF mixture was gradually incorporated in small increments to ensure even distribution. Once the dough achieved a fluffy and fine texture, transferred it to a 3D printing mold (dimensions: 15 × 6 × 4 cm^3^). The dough was frozen and shaped in the refrigerator for 40 min to achieve a firm consistency that was easy to slice. Afterward, the frozen dough was cut into slices 1.5 cm thick approximately. The slices were arranged on a baking tray and baked in an oven (K9, Guangdong Dayu Technology Industrial Co., Ltd., Foshan, China) for 20 min.

### 2.5. The Quality Properties of Soda Crackers and Cookies

#### 2.5.1. Expansion Ratio Analysis

The thickness of the soda crackers and cookies was measured using a digital thickness gauge (Evitt Henan Province Bond Technology Co., Ltd., Zhengzhou, China). Measurements were taken at the center of the samples before and after baking. Expansion ratio (X, %) was calculated using following formula:
(2)$$X\left(\%\right)=\frac{h_{2}}{h_{1}}\times 100\%$$ where h_2_ is the thickness of the sample after baking (mm), h_1_ is the thickness of the sample before baking (mm).

#### 2.5.2. Colorimetric Analysis

The color of the soda crackers and cookies was measured using a chromameter (CS-820N Hangzhou Caipu technology Co., Ltd., Hangzhou, China). The *L** (lightness, white 100/black 0), *a** (redness-greenness), and *b** (yellowness-blueness) value were recorded. Then the whiteness index (WI) [16], total color difference (ΔE), chroma (C*), and hue angle (H°) were calculated using following formula:
(3)$$WI=\sqrt{{\left(100-L*\right)}^{2}+{a*}^{2}+{b*}^{2}}$$
(4)$$\Delta E=\sqrt{{\left(L*-{L_{0}}^{*}\right)}^{2}+{\left(a^{*}-{a_{0}}^{*}\right)}^{2}+{\left(b^{*}-{b_{0}}^{*}\right)}^{2}}$$
(5)$$C^{*}=\sqrt{{a^{*}}^{2}+{b^{*}}^{2}}$$
(6)$$H{}^{\circ}=\tan^{-1}\frac{b*}{a*}$$ where *L*_0_* = 97.97, *a*_0_* = −0.42, and *b*_0_* = 0.41 represent the color parameters of the white standard plate. WI evaluates holistic whiteness, ΔE indicates deviation from the reference, C* reflects color intensity, and H° defines tonal dominance.

#### 2.5.3. The Texture of Soda Crackers and Cookies

The texture of soda crackers was measured using a texture analyzer (TA-XT plus, Shanghai Baosheng Industrial Development Co., Ltd., Shanghai, China) and the three-point bending method. The texture of the cookies was tested using the P2 probe. The test methods for both are as follows: the contact force was 0.1 N, the test speed before and after was 1 mm/s, and the downward displacement was 5 mm. The experimental type was a single test. According to the method of Mamat, et al. [17], the test parameters were peak force (N) and peak distance (mm).

### 2.6. Statistical Analysis

All experimental procedures were performed in triplicate. Quantitative analysis was executed with OriginPro 2024b (OriginLab Corporation, Northampton, MA, USA), with results expressed as mean ± standard deviation. Statistical evaluations employed one-way ANOVA coupled with Tukey’s post hoc test, maintaining a significance threshold at *p* < 0.05.

## 3. Results and Discussion

### 3.1. Pasting Properties

The pasting properties of starch, a critical determinant of food quality and texture [18], was systematically characterized through RVA profiling (Figure 1), with corresponding pasting property parameters detailed in Table 3. After CRP substitution, the peak viscosity, trough, and peak time showed a decreasing trend, while the final viscosity initially decreased and then increased. The setback showed a similar trend to the final viscosity. The breakdown initially increased and then decreased, and the pasting temperature showed no significant difference among the samples. Peak viscosity and trough gradually decreased, with the maximum values observed at C0 (376.46 ± 1.20 and 359.95 ± 0.56) and the minimum values observed at C10 (319.89 ± 2.00 and 305.13 ± 2.96), respectively. This decrease is primarily attributed to the continuous substitutions of CRP, which dilute the starch content in the mixture. At low substitution ratios (C2.5 and C5), the final viscosity was significantly lower than at C0, whereas at high substitution ratios (C7.5 and C10), the final viscosity was significantly higher than at C0. These changes can be explained by two principal mechanisms mediated by CRP supplementation. The first mechanism involves the adsorption of polysaccharide components from CRP onto starch granule surfaces, thereby forming a protective interfacial layer [19]. Meanwhile, as the protein content in the system increases significantly, proteins from *Chlamydomonas reinhardtii*, likely including both soluble and insoluble fractions, may contribute to the formation of extensive gel networks during heating and cooling [20,21], Further research is needed to identify the specific proteins involved and their individual contributions to the gel network formation. The lowest breakdown and setback were observed at C2.5, indicating the strongest shear resistance [22] and aging resistance [23]. Researchers [12] also investigated the use of *Chlorella* as a functional ingredient in cookies, observing a comparable trend in viscosity changes. Overall, based on the analysis of the starch gelatinization in the CRP-LWF composite system, the C2.5 sample appears to be a viable candidate for CRP-based baked goods.

### 3.2. The Physical Properties of Dough

#### 3.2.1. Farinograph Properties

Farinograph analysis elucidates the impact of CRP supplementation on hydration characteristics and viscoelastic behavior of dough. Farinographic curves of CRP-LWF blends were presented in Figure 2, and the key functional parameters were listed in Table 4. With the increase in CRP substitution, both the water absorption rate and degree of softening showed a gradually increasing trend, whereas the stability time exhibited a decreasing trend. The farinograph quality number at C10 was significantly higher compared to that at C0. No significant differences were observed in dough developmnet time among the samples.

Water absorption governs three critical dough system parameters: macromolecular structuring (particularly gluten matrices); the distribution and hydration of dough constituents; and overall dough yield [24]. When the proportion of CRP in the CRP-LWF composite system rose from 0% to 10%, the water absorption rate correspondingly increased from 57.2% to 63.93%. This phenomenon might be attributed to the hydrophilic properties of CRP proteins, coupled with their interaction in competing with alternative dough constituents for hydration resources [25,26]. These findings correlate with previous work by Yoon, et al. [27], who observed a 45.44% increase in the hydration capacity of the mixture (from 54.83% to 79.27%) following a 24% substitution with soy protein. The increase in degree of softening indicated a reduction in dough strength due to CRP. Nevertheless, no significant difference was observed in C2.5 compared with C0.

Stability time and farinograph quality number constitute critical metrics for assessing gluten matrix integrity and overall dough strength [28]. The stability time at C2.5 is significantly higher than at C0, which implies CRP appears to facilitate cross-linking or cohesion within gluten networks, potentially improving structural integrity [29]. The maximum farinograph quality number was observed to be 44.00 ± 3.46 in C10, suggesting that a higher proportion of CRP can significantly improve the strength and mixing tolerance of the wheat dough. Tian, et al. [30] measured the dough samples containing varying proportions of *Chlorella pyrenoidosa* powder, observing that similar changes in the farinograph quality number.

#### 3.2.2. Extensograph Properties

Extensograph analysis serves as a standard tool for characterizing dough’s viscoelastic behavior. Extensographic curves and parameters of the CRP-LWF mixtures under the proofing times of 45, 90, and 135 min are presented in Figure 3 and Table 5, respectively. Stretching energy is a critical indicator for assessing baking quality. A lower stretching energy value typically correlates with reduced baking stability of the dough. Dough made from flour with fragile solid gluten tends to exhibit lower stretching energy [31]. For short proofing times (45 min), C2.5 and C5 significantly increased the stretching energy of CRP-LWF dough, suggesting superior baking stability during shorter proofing periods. However, no significant differences were observed at longer proofing times (90 min and 135 min). Higher extensibility generally indicates greater dough resilience, while excessively low extensibility is often considered undesirable in the baking industry [32]. At 135 min proofing time, the extensibility of the C2.5 sample was significantly lower than that of C0, with the minimum value recorded as 98.67 ± 2.08 mm. No significant differences were observed between other proofing times and C0. This suggests that extended proofing times may negatively impact dough baking performance. Stretching resistance reflects the elastic properties of wheat dough, specifically its ability to resist the force applied by the measuring instrument. At 45 min and 90 min proofing times, all CRP-LWF dough samples demonstrated significantly higher stretching resistance compared to C0. At 135 min, only the C2.5 sample exhibited significantly better elastic properties relative to C0. The stretch ratio is defined as the ratio of stretching resistance to extensibility. At proofing times of 45 min, 90 min, and 135 min, the stretch ratio of the C2.5 sample was significantly higher than that of C0, reaching maximum values of 3.50 ± 0.10, 4.13 ± 0.25, and 4.40 ± 0.36, respectively. This might be because the addition of a small amount of CRP helps to form a stronger gluten network structure, which is similar to the results of farinographic properties.

### 3.3. The Quality Properties of Soda Crackers and Cookies

#### 3.3.1. Colorimetric Analysis

Color plays a critical role in consumers’ initial acceptance of baked foods. Additionally, color formation predominantly occurs during the later stages of baking [33], serving as an essential indicator for determining the completeness of the baking process. The pictures of baked soda crackers (a) and cookies (b) were presented in Figure 4. The distribution maps of *L** values (a), *a** and *b** values (b), whiteness index (WI) (c), total color difference (ΔE) (c), chroma (C*) (c), and hue angle (H°) (c) for the samples were presented in Figure 5.

The *L** value measures lightness, with *L** = 0 representing black and *L** = 100 representing white. As CRP content increased, the *L** values of soda crackers and cookies exhibited a significant downward trend, with cookies showing higher *L** values than soda crackers. This may be attributed to the higher butter content in cookies, which creates a smoother surface and enhances light reflection. The *a** value is associated with redness (+*a**) and greeness (−*a**), while the *b** value corresponds to yellowness (+*b**) and blueness (−*b**). When the CRP substitution level increased from 0% to 10%, there was a significant decrease in *b** values and an increase in *a** values. This change is primarily due to the inherent green properties of CRP combined with the Maillard reaction. The Maillard reaction, a complex series of chemical reactions between reducing sugars and amino acids, typically leads to browning and the formation of brown pigments, which are characterized by higher *b** values. However, in this study, the presence of CRP, which has a natural green hue, appears to counteract the browning effect of the Maillard reaction, resulting in a shift towards higher *a** values and lower *b** values. The ΔE values of soda crackers and cookies significantly increased with rising CRP substitution levels, indicating that different CRP substitution ratios resulted in distinguishable color differences for consumers. The overall color gradually shifts towards a more CRP-characteristic green tone, closely linked to the natural color of the raw material CRP. With increasing CRP content, the WI value progressively rised, while C* decreased, suggesting that CRP may inhibit the Maillard reaction and reduce brown substance formation. The H° of C2.5, C5, C7.5, and C10 were significantly higher than C0, resulting in an overall color shift towards green. Similar findings had been reported in many studies. Hussein, Mostafa, Ata, Hegazy, Abu-Reidah and Zaky [6] measured the color of cookies made with spirulina algae powder and found that the *a** values increased. Paula da Silva, Ferreira do Valle and Perrone [5] discovered that the b* values of vegan cookies made with Spirulina maxima biomass were significantly reduced compared to the control group.

The color changes in baked goods can sometimes enhance their appeal to consumers [34], and these changes are not necessarily negative factors affecting consumer acceptance. In this study, CRP substitution led to an increase in the *L** value and WI of soda crackers and cookies, a reduction in C*, and an overall color shift towards green. To some extent, these changes may provide a positive factor for the development of new CRP-based products.

#### 3.3.2. Expansion Ratio Analysis and the Texture of Soda Crackers and Cookies

The expansion ratio of the product is a critical factor in quality assessment, as it reflects the extent of volume increase during the baking process. The expansion radios analysis were presented in Table 6. The expansion ratio of the sample significantly decreased when the substitution ratio of CRP in LWF exceed 2.5%. The expansion of soda crackers is primarily attributed to the addition of yeast, and cookies results from syrup formation in the dough during baking. Specifically, the sugar added during the production process combines with water and undergoes caramelization at high temperatures to form syrup, upon incorporating CRP, the reduced water availability leads to less syrup formation, thereby restraining samples expansion [13]. The higher expansion ratio of soda crackers compared to cookies may be attributed to the addition of yeast during production, which enhances gas generation and promotes volume expansion.

Texture is a critical attribute of food quality and plays a pivotal role in shaping consumers’ overall quality perception of food products. Together with appearance and taste, texture constitutes one of the three primary factors that influence consumer acceptance of food, thereby determining its overall acceptability [35]. The texture analysis of soda crackers and cookies are presented in Table 6. The peak force measured during the first compression cycle serves as an indicator of product hardness [36]. When the CRP substitution ratio exceeded 2.5%, the peak force of soda crackers was significantly lower than C0. Similarly, the peak force of cookies also decreased significantly compared to C0 when the CRP substitution ratio reached 10%. This phenomenon may be attributed to the increasing proportion of CRP in the CRP-LWF system, which enhanced its dilution effect on the gluten network; the gas retention capacity was weakened, and the air cell structure became less developed [37]. The peak distance is typically associated with the of the product. The peak distances of C5, C7.5, and C10 are significantly lower than that of C0, suggesting that CRP could substantially influence the brittleness of baked goods. However, this effect may not be desirable [38]. Additionally, soda crackers exhibited higher hardness and brittleness compared to cookies, which may result from the synergistic effects of sugar and butter; the fat content could affect the crystallization behavior and distribution of sugar after baking [17,39,40]. Hussein, Mostafa, Ata, Hegazy, Abu-Reidah and Zaky [6] found that as the amount of spirulina algae powder added increased, the hardness of the biscuits decreased, and Lu, He, Liu, Wen and Xia [14]. reported that the hardness and brittleness of biscuits increased with the addition of chickpea flour.

Overall, these findings are in agreement with the pasting properties of CRP-LWF and the physical properties of the dough. Specifically, at a CRP substitution level of C2.5, both the dough characteristics and baked product quality achieved their peak performance.

## 4. Conclusions

This study revealed that incorporating CRP into flour-based systems can significantly influence the pasting properties of CRP-LWF mixed flour, the physical properties of dough, and the quality of soda crackers and cookies. The C2.5 sample demonstrated superior pasting properties, farinographic characteristics, and extensographic performance. Research into the quality properties of soda crackers and cookies further corroborated this finding: at a CRP substitution rate of 2.5%, both dough characteristics and baked product quality achieved optimal levels. The incorporation of CRP resulted in an elevated *L** value and WI for soda crackers and cookies, a diminished C*, and a perceptible shift in overall color toward greener tones. Overall, this study offers a comprehensive exploration for baked foods enriched with CRP. Nevertheless, further research is required to validate its nutritional value and to develop appropriate pretreatment methods for the raw materials, thereby mitigating the adverse effects of CRP’s color and distinctive flavor on the quality of baked products.

## Figures and Tables

**Figure 1 foods-14-03621-f001:**
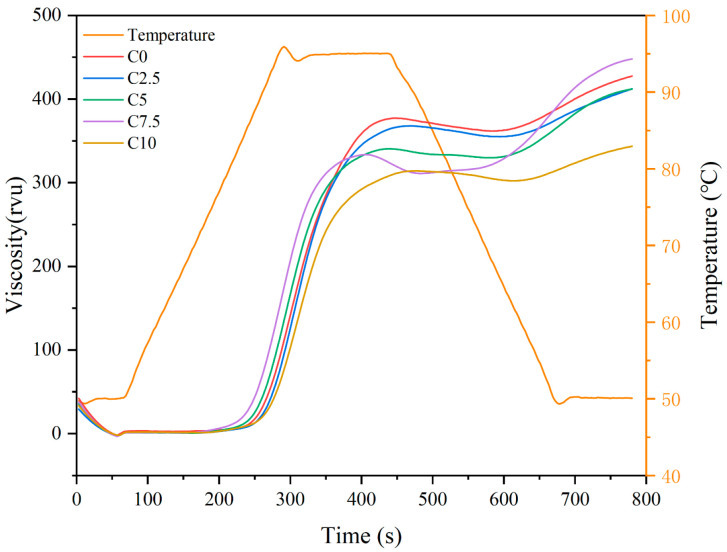
RVA curves of low-gluten wheat flour blends containing varying concentrations of *Chlamydomonas reinhardtii* powder. The experimental groups (C0–C10) correspond to *Chlamydomonas reinhardtii* powder supplementation levels of 0–10% (*w*/*w*).

**Figure 2 foods-14-03621-f002:**
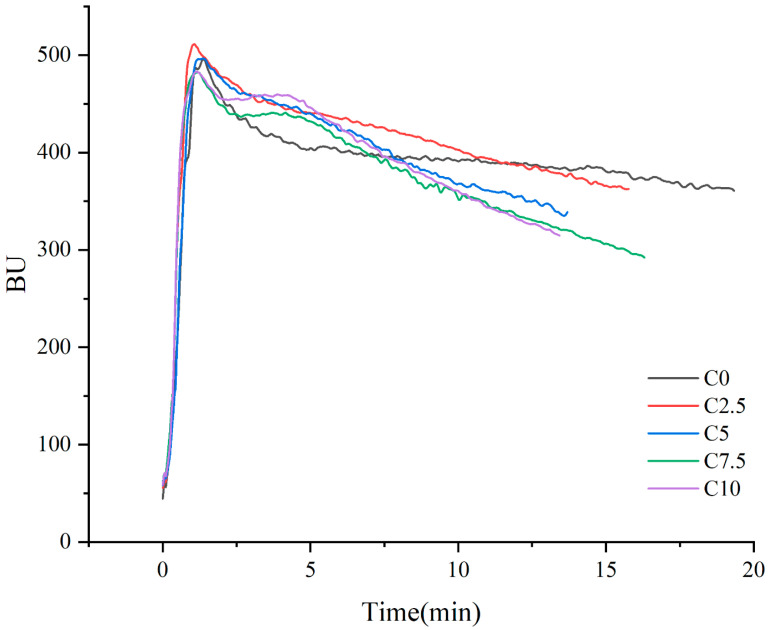
Farinograph curves of low-gluten wheat dough blends containing varying concentrations of *Chlamydomonas reinhardtii* powder. The experimental groups (C0–C10) correspond to *Chlamydomonas reinhardtii* powder supplementation levels of 0–10% (*w*/*w*)**.** BU, Brabender unit.

**Figure 3 foods-14-03621-f003:**
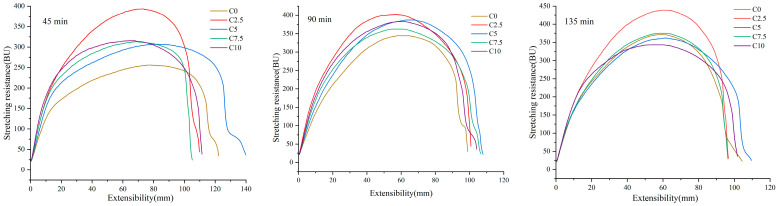
Extensograph curves of low-gluten wheat dough blends containing varying concentrations of *Chlamydomonas reinhardtii* powder. The experimental groups (C0–C10) correspond to *Chlamydomonas reinhardtii* powder supplementation levels of 0–10% (*w*/*w*), respectively.

**Figure 4 foods-14-03621-f004:**
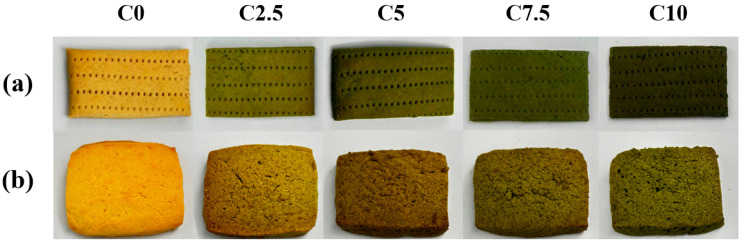
The picture of baked soda crackers (**a**) and cookies (**b**). Note: The experimental groups (C0–C10) correspond to *Chlamydomonas reinhardtii* powder supplementation levels of 0–10% (*w*/*w*).

**Figure 5 foods-14-03621-f005:**
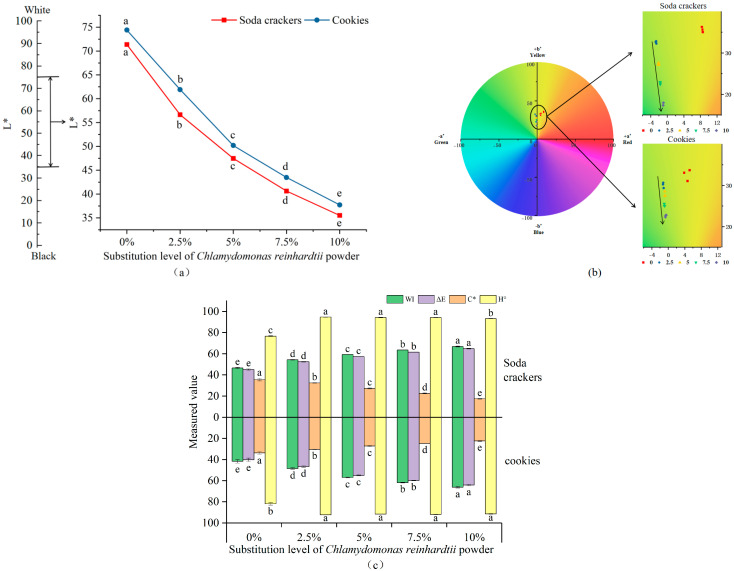
Color analysis diagram of the sample. (**a**) The *L** values of sample; (**b**) Distribution chart of *a** and *b** values; (**c**) The whiteness index (WI), total color difference (ΔE), chroma (C*), and hue angle (H°) values of sample. Values with different superscripts are significantly different (*p* < 0.05).

**Table 1 foods-14-03621-t001:** Nutritional composition of low-gluten wheat flour and *Chlamydomonas reinhardtii* powder mixtures at varying replacement ratios, where CRP represents *Chlamydomonas reinhardtii* powder. LWF represents low-gluten wheat flour.

CRP (%)	LWF (%)	Protein (g/100 g)	Fat (g/100 g)	Carbohydrate (g/100 g)	Moisture (g/100 g)	Ash (g/100 g)
0	100	8.50	1.10	76.40	13.00	0.45
2.5	97.5	9.38	1.19	75.52	12.75	0.62
5	95	10.26	1.28	74.66	12.50	0.78
7.5	92.5	11.13	1.37	73.79	12.24	0.95
10	90	12.01	1.46	72.93	11.99	1.12

**Table 2 foods-14-03621-t002:** Formulation and process parameters for incorporating *Chlamydomonas reinhardtii* Powder into soda crackers and cookies, where C0-C10 represent samples with 0–10% CRP content. CRP represents *Chlamydomonas reinhardtii* powder.

		Soda Crackers					Cookies				
		C0	C2.5	C5	C7.5	C10	C0	C2.5	C5	C7.5	C10
**Formulation**	CRP (g)	0	3.75	7.5	11.25	15	0	3.75	7.5	11.25	15
	Flour (g)	150	146.25	142.5	138.75	135	150	146.25	142.5	138.75	135
	Salt (g)	2					1				
	Butter (g)	30					100				
	Sugar (g)	-					60				
	Yeast (g)	3					-				
	Milk (g)	60					-				
	Egg	-					1				
	Baking soda (g)	1					-				
**Key** **Process Parameters**	Fermentation	25 °C, 75% humidity, 30 min					-				
	Refrigeration time (min)	-					40				
	Baking Parameters	Top: 175 °C, Bottom: 155 °C, 20 min					Top: 175 °C, Bottom: 155 °C, 20 min				

**Table 3 foods-14-03621-t003:** The pasting properties of the mixture of low-gluten wheat flour and *Chlamydomonas reinhardtii* powder.

	Pasting Temperature (°C)	Peak Viscosity (rvu)	Peak Time (min)	Trough (rvu)	Final Viscosity (rvu)	Breakdown (rvu)	Setback (rvu)
C0	80.63 ± 1.34 ^a^	376.46 ± 1.20 ^a^	7.20 ± 0.00 ^a^	359.95 ± 0.56 ^a^	429.25 ± 0.72 ^b^	16.51 ± 0.64 ^c^	69.31 ± 1.28 ^c^
C2.5	80.80 ± 1.94 ^a^	361.49 ± 2.55 ^b^	7.20 ± 0.00 ^a^	349.83 ± 3.19 ^a^	411.39 ± 1.71 ^c^	11.67 ± 0.64 ^d^	61.57 ± 1.48 ^d^
C5	77.68 ± 2.52 ^a^	355.09 ± 3.62 ^b^	7.12 ± 0.10 ^ab^	334.39 ± 3.74 ^b^	410.35 ± 3.51 ^c^	20.71 ± 0.12 ^b^	75.96 ± 0.23 ^b^
C7.5	75.51 ± 2.07 ^a^	337.07 ± 2.87 ^c^	6.92 ± 0.18 ^b^	309.46 ± 1.12 ^c^	449.46 ± 0.68 ^a^	27.61 ± 1.75 ^a^	140.00 ± 0.44 ^a^
C10	80.15 ± 3.22 ^a^	319.89 ± 2.00 ^d^	7.15 ± 0.09 ^ab^	305.13 ± 2.96 ^c^	444.59 ± 4.19 ^a^	14.77 ± 0.95 ^cd^	139.45 ± 1.22 ^a^

Note: The experimental groups (C0–C10) correspond to *Chlamydomonas reinhardtii* powder supplementation levels of 0–10% (*w*/*w*). Within each column, values with different superscripts are significantly different (*p* < 0.05).

**Table 4 foods-14-03621-t004:** The farinographic properties of the mixed dough of low-gluten wheat flour and *Chlamydomonas reinhardtii* powder.

	Water Absorption (%)	Dough Development Time (min)	Stability Time (min)	Degree of Softening (FU)	Farinograph Quality Number
C0	57.20 ± 0.10 ^d^	1.27 ± 0.12 ^a^	1.17 ± 0.06 ^b^	114.67 ± 7.64 ^b^	18.33 ± 0.58 ^b^
C2.5	59.10 ± 0.30 ^c^	1.10 ± 0.10 ^a^	1.47 ± 0.15 ^a^	124.00 ± 5.57 ^b^	21.33 ± 3.21 ^b^
C5	61.17 ± 0.21 ^b^	1.13 ± 0.15 ^a^	1.17 ± 0.12 ^b^	157.33 ± 4.51 ^a^	21.00 ± 2.00 ^b^
C7.5	61.93 ± 0.25 ^b^	1.07 ± 0.15 ^a^	0.83 ± 0.06 ^c^	159.00 ± 1.00 ^a^	17.67 ± 0.58 ^b^
C10	63.93 ± 0.71 ^a^	1.07 ± 0.15 ^a^	0.73 ± 0.06 ^c^	166.67 ± 5.51 ^a^	44.00 ± 3.46 ^a^

Note: The experimental groups (C0–C10) correspond to *Chlamydomonas reinhardtii* powder supplementation levels of 0–10% (*w*/*w*). Within each column, values with different superscripts are significantly different (*p* < 0.05).

**Table 5 foods-14-03621-t005:** The extensographic properties of the mixed dough of low-gluten wheat flour and *Chlamydomonas reinhardtii* powder.

Proofing Time (min)	Sample	Stretching Energy (cm^2^)	Extensibility (mm)	Stretching Resistance (BU)	Stretch Ratio
45	C0	43.00 ± 3.00 ^c^	121.67 ± 3.21 ^a^	265.00 ± 5.57 ^c^	2.17 ± 0.06 ^c^
	C2.5	59.33 ± 1.53 ^a^	111.00 ± 1.73 ^a^	390.33 ± 10.26 ^a^	3.50 ± 0.10 ^a^
	C5	53.33 ± 4.93 ^ab^	129.67 ± 13.05 ^a^	304.00 ± 14.73 ^b^	2.37 ± 0.29 ^bc^
	C7.5	48.33 ± 1.53 ^bc^	114.00 ± 11.36 ^a^	311.67 ± 9.07 ^b^	2.77 ± 0.32 ^b^
	C10	50.33 ± 3.06 ^bc^	114.00 ± 2.00 ^a^	316.00 ± 17.00 ^b^	2.77 ± 0.15 ^b^
90	C0	46.67 ± 2.08 ^a^	107.33 ± 6.66 ^a^	334.33 ± 9.07 ^c^	3.13 ± 0.25 ^b^
	C2.5	53.67 ± 2.31 ^a^	99.00 ± 3.61 ^a^	410.33 ± 20.50 ^a^	4.13 ± 0.25 ^a^
	C5	53.33 ± 2.52 ^a^	105.67 ± 6.66 ^a^	380.33 ± 11.55 ^ab^	3.60 ± 0.30 ^ab^
	C7.5	51.33 ± 3.06 ^a^	104.67 ± 4.93 ^a^	361.33 ± 11.59 ^bc^	3.43 ± 0.06 ^ab^
	C10	53.67 ± 3.06 ^a^	103.00 ± 5.29 ^a^	390.00 ± 11.27 ^ab^	3.80 ± 0.35 ^ab^
135	C0	49.33 ± 3.21 ^ab^	110.00 ± 4.36 ^a^	369.33 ± 25.54 ^ab^	3.37 ± 0.23 ^b^
	C2.5	57.33 ± 3.06 ^a^	98.67 ± 2.08 ^b^	413.67 ± 21.94 ^a^	4.40 ± 0.36 ^a^
	C5	51.67 ± 2.52 ^ab^	104.33 ± 4.93 ^ab^	371.33 ± 20.65 ^ab^	3.57 ± 0.31 ^b^
	C7.5	49.67 ± 5.03 ^ab^	99.67 ± 5.51 ^ab^	370.00 ± 18.03 ^ab^	3.70 ± 0.17 ^b^
	C10	48.33 ± 2.08 ^b^	100.33 ± 2.89 ^ab^	350.33 ± 12.70 ^b^	3.50 ± 0.10 ^b^

Note: The experimental groups (C0–C10) correspond to *Chlamydomonas reinhardtii* powder supplementation levels of 0–10% (*w*/*w*). Within each column, values with different superscripts are significantly different (*p* < 0.05).

**Table 6 foods-14-03621-t006:** Impact of graded *Chlamydomonas reinhardtii* powder incorporation on expansion ratio and textural properties.

	Sample	Expansion Ratio (%)	Peak Force (N)	Peak Distance (mm)
Soda crackers	C0	1.89 ± 0.10 ^a^	24.38 ± 0.59 ^b^	39.11 ± 0.01 ^ab^
	C2.5	1.74 ± 0.02 ^ab^	25.47 ± 2.35 ^b^	38.38 ± 1.25 ^b^
	C5	1.65 ± 0.02 ^b^	31.50 ± 0.78 ^a^	40.32 ± 0.34 ^a^
	C7.5	1.49 ± 0.09 ^c^	32.97 ± 2.26 ^a^	39.54 ± 0.14 ^ab^
	C10	1.41 ± 0.01 ^c^	31.70 ± 0.32 ^a^	40.46 ± 0.38 ^a^
Cookies	C0	1.26 ± 0.01 ^a^	3.70 ± 0.27 ^b^	11.69 ± 0.33 ^a^
	C2.5	1.21 ± 0.03 ^ab^	3.60 ± 0.04 ^b^	11.31 ± 0.98 ^ab^
	C5	1.16 ± 0.05 ^bc^	4.25 ± 0.52 ^b^	10.21 ± 0.15 ^bc^
	C7.5	1.13 ± 0.00 ^c^	4.40 ± 0.09 ^b^	10.19 ± 0.24 ^bc^
	C10	1.09 ± 0.02 ^c^	5.38 ± 0.35 ^a^	9.47 ± 0.07 ^c^

Note: The experimental groups (C0–C10) correspond to *Chlamydomonas reinhardtii* powder supplementation levels of 0–10% (*w*/*w*). Within each column, values with different superscripts are significantly different (*p* < 0.05).

## Data Availability

The original contributions presented in this study are included in the article. Further inquiries can be directed to the corresponding author.

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
