# Peer review of "Sustainable Incorporation of Chlamydomonas reinhardtii Powder into Flour-Based Systems: Investigating Its Influence on Flour Pasting Properties, Dough Physical Properties, and Baked Product Quality"

_foods, 2025, doi:10.3390/foods14213621_

Round 1

Reviewer 1 Report

Comments and Suggestions for Authors

Reference: foods-3922416-peer-review-v1

Title: Sustainable Incorporation of Chlamydomonas reinhardtii Powder into Flour-Based Systems: Investigating its Influence on Flour Pasting Properties, Dough Physical Properties, and Baked Product Quality

Journal: Foods

The manuscript Sustainable Incorporation of Chlamydomonas reinhardtii Powder into Flour-Based Systems: Investigating its Influence on Flour Pasting Properties, Dough Physical Properties, and Baked Product Quality focuses on the incorporation of  Chlamydomonas reinhardtii powder into flour doughs to produce enriched soda crackers and cookies. The manuscript presents major concerns:

  • First, the writing is not fluid and contains many linguistic and interpretation errors.
  • Second, the experimental approach and analyses performed are insufficient for publication in a journal such as Foods.

Comments and suggestions to the authors are given below.

Abstract

  • Lines 17-19: “Then among the expansion ratios of soda crackers and cookies, C2.5 achieved the highest value with no significant difference compared to C0”: This sentence contains a contradiction. Please correct it.
  • Lines 19-20: “…the texture characteristics of C2.5 were superior”: superior to what and in what way?

Introduction

  • Lines 73-75: The authors claim that “This study provides a robust theoretical foundation for the market development of CRP-enriched soda crackers and cookies” which is wrong and exaggerated in the absence of instrumental rheological measurements of dough viscoelastic properties, and also in the absence of sensory analyses of the finished products. To be corrected.

Materials and Methods

  • Line 107: If the authors used Brabender Farinograph, il should be specified in the text.
  • Line 117: Same remark for the Brabender Extensograph.
  • Line 121: what was measured with the Extensograph, at which “constant extension rate”?
  • The meaning of the parameters measured in Table 4 and the Stretch ratio equation must be provided.
  • Line 127: Replace “were presented in Table 1” by “are presented in Table 1”.
  • Line 144: What was the “dough sheeter” used to flatten the dough into a uniform sheet? How was it used to ensure the results’ reliability?
  • Line 147: What was the temperature of the oven? Why “baked in an oven for 15 min” whereas in Table 1 it is indicated 20 min?
  • Line 155: Add (X, %) after “Expansion ratio”.
  • Line 158: Remove “X represents the expansion ratio (%)”.
  • Line 177: give the “contact force” in N (Newton).
  • Line 179: No capital letters in “Peak force (N) and Peak distance (mm)”.

Results and discussion

  • Lines 190-191: Where is the increasing trend in: “…while the final viscosity and setback demonstrated an increasing trend”?
  • Line 203: Explain how “proteins form extensive gel networks during heating and cooling” essentially proteins from Chlamydomonas reinhardtii, which type of proteins are involved here? Soluble or insoluble? Are they thermally coagulable? Give references.
  • Line 224: “…showed significant increased”: To be corrected.
  • Line 238: What is “farinograph quality number”? Give the equation in Materials and Methods section.
  • Line 245: “… observing that a consistent change in …”: To be corrected.
  • Line 257: “The extensograph properties are commonly utilized instruments…”: To be corrected.
  • Line 283: “…incorporating 3% carob fibre…”: The comparison is not valid! Carob fiber is very different from Chlamydomonas reinhardtii
  • Figure 3: Complete the caption: What are mm and EU?
  • Line 302: Correct the sentence: “The picture … were presented in Figure 4.
  • Line 309: Explain how “This may be attributed to the higher butter content in cookies”.
  • Line 314 and 320: The contribution of “Maillard reaction” is not properly explained.
  • Lines 322-323: Explain how “Similar findings had been reported in the studies by…”. On which products? What were their results briefly.
  • Line 328: “…these changes may enhance consumer attraction”: How can you assert or assume this without any reliable sensory analysis? To be corrected.
  • Line 342: Replace “analysis was presented in Table 5” by “analysis is presented in Table 5.
  • Lines 343-344: “The expansion of soda crackers and cookies results from syrup formation in the dough during baking”: What is this syrup and where does it come from? This is not clear!
  • Line 353: Replace “…were presented in Table 5” by “…are presented in Table 5”.
  • Lines 363-364: How can you assert that: “…soda crackers exhibited better textural properties compared to cookies”?
  • Line 366: Explain which type of “Comparable findings…”?
  • Line 390: It's an exaggeration to say: “…solid theoretical foundation”! To be removed.

Reviewer 2 Report

Comments and Suggestions for Authors

The research is very interesting, please address the comments to strengthen the content.
